# SNP and Haplotype Variability in the *BnP5CR2* Gene and Association with Resistance and Susceptible Cultivars for *Sclerotinia sclerotiorum* in *Brassica napus*

**Yu Zhang** [1,2,*], **Yu Wang** [1], **Dong Wu** [1], **Dong Qu** [1,3], **Xiaomin Sun** [4] **and Xiaojuan Zhang** [1,5]

1    Shaanxi University of Technology, Hanzhong 723000, China
2    Qinba State Key Laboratory of Biological Resources and Ecological Environment, Hanzhong 723000, China
3    Shaanxi Province Key Laboratory of Bio-resources, Hanzhong 723000, China
4    Hanzhong Agricultural Sciences Institute, Hanzhong 723000, China
5    QinLing-Bashan Mountains Bioresources Comprehensive Development C. I. C., Hanzhong 723000, China
*    Correspondence: yuzhang20160315@outlook.com

**Abstract:** *Sclerotinia sclerotiorum* is a serious disease of oil crop. The *P5CR* gene is the first gene reported to be associated with resistance to *Sclerotinia* infection in soybeans, and its closest homologs are located on chromosomes A10 and C09 of *Brassica napus*. We named these *BnP5CR1* and *BnP5CR2*, respectively. The purpose of this study was to examine the single-nucleotide polymorphism (SNP) and haplotype diversity (Hd) of *BnP5CR2* among canola cultivars with different levels of resistance to *S. sclerotiorum* as well as the expression patterns of *BnP5CR2* via an association analysis using resistant and susceptible cultivars of *B. napus*. The results can thus provide information for future research on the mechanisms of disease resistance to *S. sclerotiorum* and the breeding of resistant canola cultivars. A total of 95 and 12 polymorphic sites were detected in 1870 and 678 SNP sites in 16 *BnP5CR2* and their coding DNA sequence (CDS) population, respectively. A total of six different haplotypes (H1–H6) were inferred from the 16 *BnP5CR2* gene-CDS that contributed to the high level of polymorphism. Hd was equal to 0.617, and H1 shared by 10 cultivars was the dominant haplotype, suggesting that H1 is an ancient haplotype among the *BnP5CR2* genes. H6 and H5 haplotpypes were present in Nan12R and ZhongYou821, respectively. The expression level in vitro of the *BnP5CR2* between Nan12R and ZhongYou821 was significantly different. The upregulated expression of *BnP5CR2* in resistant cultivars was higher than that of susceptible cultivars under 6 h, 12 h, 24 h, and 36 h treatments of pathogen stress, among which the expression level was significantly increased at 6 h, 12 h and 36 h in resistant cultivars, and the difference reached a highly significant level at 6 h ($p < 0.01$). The two cultivars with clear differences in expression features possessed different *BnP5CR2* gene-CDS-haplotypes, indicating that gene-CDS-haplotype diversity may have greater power than SNPs for the detection of causal genes for quantitative traits.

**Keywords:** *Brassica napus*; *BnP5CR2*; cloning; SNP; haplotype; expression features





## 1. Introduction

*Sclerotinia sclerotiorum* (Lib.) de Bary is a notorious fungal pathogen with worldwide distribution that can infect many important crops, such as canola, soybean, sunflower and cause *Sclerotinia* diseases, and thus is a serious problem requiring to be solved urgently in agricultural production. The most effective way to control *S. sclerotiorum* of canola is to seek the resistant source of its host and cultivate resistant cultivars and clone the dominant resistant genes. In 2021, it was reported for the first time that the *P5CR* (pyrroline-5-carboxylate reductase) gene was associated with a significant resistance to *S. sclerotiorum* infection in soybean [1]. *P5CR* is an important housekeeping gene of eukaryotes. P5C (Δ1-pyrroline-5-carboxylate)—the intermediate metabolite of proline, ornithine, arginine, and glutamic acid—is reduced to proline by *P5CR*, which is the final step in the biosynthesis

of proline. In plants, proline not only constitutes an essential amino acid and cell wall synthesis but is also a type of osmotic stress regulator, playing a pivotal role in controlling the osmotic stress imposed on plants, especially under dry and salty environments; as such, proline accumulation in plants as well as its related gene expression have continually been the subject of numerous studies [2–9]. Experiments showed that knocking-out *P5CR* in *Arabidopsis thaliana* resulted in embryonic death of the plants [10]. A few reports exist that pertain to the cloning and polymorphism of the *P5CR* gene in plants: in 1990, the soybean *P5CR* gene was cloned successfully [11]; in 1992, the *P5CR* gene was isolated from peas [12]; in 1998, the *P5CS* and *P5CR* genes were cloned from kiwi and have showed, via Northern hybridization, that two genes are involved in transcription regulation [13]; in 2013, the *P5CS* and *P5CR* genes were cloned from *indica rice* [14]; in 2015, researchers used the RACE technique to clone the *P5CR* gene in *Lolium perenne*, the full length of the gene—which responds to various environmental stressors—being 1047 bp, with a 94.3% similarity with the predicted gene [15]; in 2016, researchers cloned the *P5CR* gene from *Manihot esculenta*, the full length of the associated cDNA being 828 bp [16]; in 2017, researchers cloned the *P5CR* gene of *Tamarix hispida* and analyzed its expression in high salt environment [17]. However, limited information is available on enzymes related to proline accumulation under pathogen stress, and the relationship between *P5CR* diversity and the resistance to *S. sclerotiorum* as well as the expression features of the *P5CR* after Sclerotiorum infection between resistance and susceptible canola cultivars. *P5CR* is a superfamily of genes that contains at least eight members in *B. napus*. The *P5CR* reported in soybeans was aligned to loci on chromosomes A10 and C09 of *B. napus* as the closest homologs, and we named these *BnP5CR1* and *BnP5CR2*, respectively. The SNP polymorphism of *BnP5CR1* was not associated with Sclerotinia resistance in *B. napus* [18]. It has been reported that SNP polymorphism within the CDS represents the most important part of the overall genetic diversity of rice, and thus may be a functional polymorphism. Some studies have shown that haplotype has a greater effect on plant disease resistance than a single SNP site [19]. Therefore, the present study focused on the SNPs and haplotype diversity of *BnP5CR2* and the expression patterns of *BnP5CR2* among cultivars of *B. napus* resistant and susceptible to *S. sclerotiorum*. The goal was to provide important information for further study of resistance mechanisms of canola as well as a reference for improving the resistance to *S. sclerotiorum* in the breeding of *B. napus*.

## 2. Materials and Methods

### 2.1. Isolation and Identification of the Pathogen of Sclerotinia sclerotiorum in Brassica napus

A total of 15 cultivars of *Brassica napus* that were representative of different phenotypes in response to *S. sclerotiorum* infection were collected from the Hanzhong Agricultural Sciences Institute (Hanzhong, China). The pathogens used in this study were collected during 2017–2018 year from the stalks of naturally diseased *B. napus,* which were identified and purified by Xiaomin Sun of Hanzhong Agriculture Science Institute. These strains were activated seven to eight days prior to inoculation.

### 2.2. Field Experiments on Identification of Disease Resistance

Canola seeds were planted at the field site of the Hanzhong Agriculture Science Institute (E: 106°59′57″, N: 33°7′48″) during three consecutive years (2019, 2020, 2021), with planting dates of 17 October 2019, 18 October 2020, and 20 October 2021, according to a 45 cm × 20 cm split-split-plot design. About ten seedlings of each cultivar were retained. Toothpicks with uniform thickness were chosen and boiled for 5 to 10 min in a 5% sucrose solution, after being placed in culture bottles for sterilization. Mycelia were inoculated in the center of the bottle and transferred into the incubator for culture at 25 °C for 6–7 days so as to make the toothpicks full of mycelium ("Local Standard of PSJG 1107.1-2009"). At the beginning of the canola flowering, the toothpick with mycelium was directly inserted into the angle between the main branch and the branch where the angle is 25–30 cm above the ground, then it was covered with a wet cotton pad to keep

it moist. Water was sprayed once in the morning and once in the evening for one week. Subsequent to continuous inoculation and observation for disease resistance for three years. The relative disease rating investigation and assessment standard are based on "Local Standard of DB51T 1035-2010 rule of field-resistance identification for *Sclerotinia sclerotiorum* (Lib.) de Bary on oilseed rape", in which the HR, MR, R, S, MS, HS represented high resistance, moderate resistance, resistance, susceptibility, moderate susceptibility and high susceptibility, respectively.

### 2.3. Genomic DNA Extraction, Primer Synthesis, and PCR Amplification

Fresh young leaves of each cultivar were collected and ground with liquid nitrogen into a powder, which was used to extract genomic DNA via the CTAB (Cetyltrimethyl ammonium bromide) method [20] and detected with 0.8% agarose gel. The primers were applied to amplify the *BnP5CR2* gene (Table 1), which were synthesized by Beijing Aoke Biotechnology Co., Ltd. (Beijing, China). The PCR (polymerase chain reaction) amplification products were separated using 2.0% agarose gel and purified using Biospin gel extraction kit (Xi'an Baiaolaibo Biotechnology Co., Ltd., Xi'an, China).

**Table 1.** The primers information of *BnP5CR2* gene.

| Gene | Primer Sequence (5′–3′) | Annealing Temperature (°C) |
|---|---|---|
| *BnP5CR2* | Forward primer 1: CAGTCAATAATTTATTTTGCGATGG; Reverse primer 1: AACTGCAGCTGGTGTATTAG. | 56 |
| | Forward primer 2: TCAGTGTTTGCATCTAACATGCTCC; Reverse primer 2: TAACAAAGTAGCTGAGATCTGAACC. | 60 |

### 2.4. Ligation and Sequencing

The purified PCR products were ligated into the pMD19-T vector, and following cloning, Sanger sequencing and sequence analysis, they were designated as *BnP5CR2*. sequencing was undertaken by Beijing Aoke Biotechnology Co., Ltd. (Beijing, China).

### 2.5. SNP Polymorphism of BnP5CR2

MEGA X [21] was used to analyze SNP polymorphism including the gene and its CDS, in which the position information of 15 CDS of *BnP5CR2* gene was obtained and saved according to the CDS information of ZS11 cultivar (https://www.ncbi.nlm.nih.gov/nuccore/NC_027775.2?from=45916942&to=45918629&report=genbank) (accessed on 10 August 2022). ZS11 is a cultivar of *B. napus* with representative genome.

### 2.6. Evolutionary Relationships Analysis of BnP5CR2

Evolutionary relationships analyses were conducted in MEGA X, in which the evolutionary history was inferred by using the Neighbor-Joining method with Bootstrap 5000 times and Tamura-Nei model [22]. Initial trees for the heuristic search were obtained automatically by applying Neighbor-Join and BioNJ algorithms to a matrix of pairwise distances estimated using the Tamura-Nei model, and then selecting the topology with superior log likelihood value.

### 2.7. BnP5CR2-CDS-Haplotype Diversity and Network Analysis

Haplotype diversity analyses were conducted in MEGA X and DnaSPV6.12.03 (http://www.ub.edu/dnasp/) (accessed on 12 August 2022) [23], the evolutionary history was inferred using the Neighbor-Joining method with Bootstrap 500 times and the Tamura-Nei model. Haplotype network construction was carried out by using MP by Network [24]. Ancestral states were inferred using the Neighbor-Joining method [25] and the Tamura-Nei

model. The trees show a set of possible nucleotides (states) at each ancestral node based on their inferred likelihood at site 1. The set of states at each node is ordered from most likely to least likely, excluding states with probabilities below 5%. The initial tree was inferred using the method. The rates among sites were treated as being uniform among sites (Uniform rates option).

*2.8. Expression Features Analysis of the BnP5CR2*

Function leaves with a diameter of about 5–6 cm were collected, washed with distilled water, and the stalks of the leaves were wrapped with moist absorbent cotton, putting them in a foam box with ice packs, and transporting them back to the laboratory, then placing them in a Petri dish with sterile filter paper and an appropriate amount of distilled water at the bottom of the dish. *S. sclerotiorum* cultured on potato dextrose agar medium was taken from round clumps of appropriate size and inoculated onto the leaves [26]. The leaves were then incubated in an incubator maintained at a relative humidity of 100% and 28 °C. The leaves which were infected were chosen at 0 h, 6 h, 12 h, 24 h, 36 h under pathogen stress, put into a cryotube and stored at −80 °C for future study. Total RNA was extracted from the 0 h, 6 h, 12 h, 24 h, 36 h under pathogen stress sample by the methods of TRIzol Reagent, Invitrogen. A PrimeScript™ RT reagent kit (Perfect Real Time; Takara Biotechnology Co., Ltd., Dalian, China) was used to reverse and transcribe the total RNA. According to the sequence of cDNA of *BnP5CR2* in NCBI, the primers were designed by Premier 5.0 software. *UBC21* was used as the internal reference gene for RT-qPCR in *B. napus* [27] (Table 2). The RT-qPCR was performed on StepOne™ (Applied Biosystems, Waltham, MA, USA), Roche FastStart™ Universal SYBR® Green premix (ROX) was used, which protocol consisted of an initial heat activation step of 95 °C for 10 min, followed by 40 cycles of 95 C° for 15 s and 60 C° for 40 s [28], and data acquisition by stepone software V2.3. Three biological replicates were performed for each treatment, and each biological replicate consisted of three technical replicates. IBM SPSS Statistics, which applies a LSD algorithm, was used to assess significant difference of the mRNA relative expression level of *BnP5CR2* between two treatments [29], drawing diagrams with R.

**Table 2.** The primers of target gene and internal reference gene for RT-qPCR.

| Gene Name | NCBI Accession No. | Forward Primer Sequence (5′–3′) | Reverse Primers Sequence (5′–3′) |
|---|---|---|---|
| *BnP5CR2* | Gene ID: 106416421 | GCTAATCGAAGCCGTGAACTC | AGTTTTCACATCTAC-CAAAGCAATA |
| *UBC21* | Gene ID: 832645 | CCTCTGCAGCCTCCTCAAGT | CATATCTCCCCTGTCT-TGAAATGC |

## 3. Results

*3.1. Identification of Sclerotinia sclerotiorum Resistance in Brassica napus*

A total of 15 cultivars were selected representing disease resistant and disease sensitive cultivars of phenotypic records (Figure 1), then to assess resistance rating (Table 3).

**Table 3.** The information of disease rating of *Sclerotinia sclerotiorum* infection.

| Sample | Disease Index | Relative Resistance Index | Resistance Rating | Sample | Disease Index | Relative Resistance Index | Resistance Rating |
|---|---|---|---|---|---|---|---|
| Hu16 | 17.14 | −0.432 | R | QingCaiR | 27.72 | 0.186 | S |
| ShanYou 28 | 22.02 | −0.121 | R | ShanYou 107 | 50.69 | 1.172 | MS |
| Zhong 11C | 21.66 | −0.142 | R | ChuanYou 36 | 55.48 | 1.364 | MS |
| QinYou 28 | 23.44 | −0.040 | R | GuoHaoYou 5 | 60.64 | 1.576 | MS |
| ZhongYou 821 | 24.16 | 0.000 | R | ChuanZa | 71.25 | 2.052 | HS |
| ZiJingYouCai | 24.87 | 0.038 | S | Nan12R | 85.16 | 2.891 | HS |
| HanYou 8 | 24.53 | 0.020 | S | CIR | 88.15 | 3.151 | HS |
| NQ2107 | 26.51 | 0.124 | S | | | | |

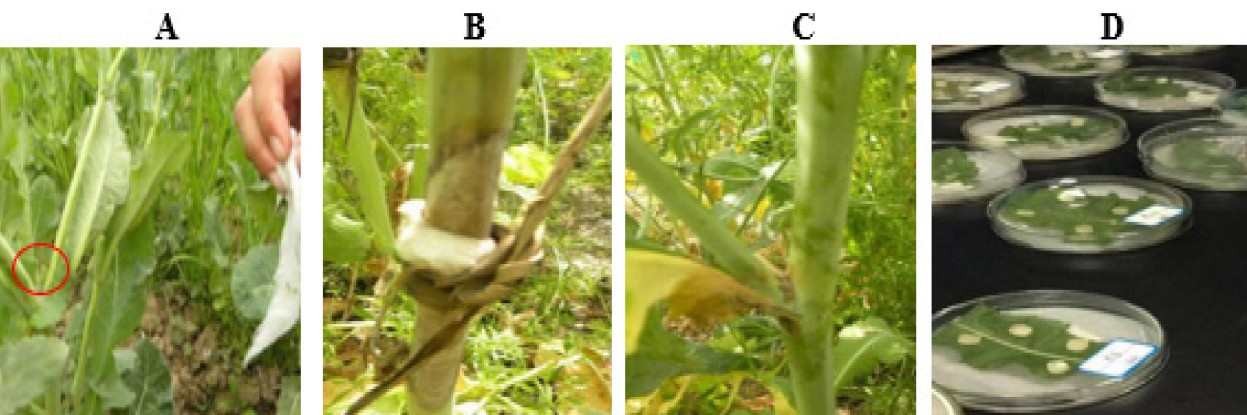

**Figure 1.** Toothpicks inoculation (red circle). (**A**) Toothpicks inoculation. (**B**) Symptoms of *Sclerotinia sclerotiorum* stem rot (14 d after inoculation) on stems of susceptible cultivars inoculated with the toothpicks method. (**C**) Symptoms of *Sclerotinia sclerotiorum* stem rot (14 d after inoculation) on stems of resistant cultivars inoculated with the toothpicks method. (**D**) Inoculation on detached leaves.

*3.2. SNP Polymorphisms of BnP5CR2*

There were 95 polymorphic sites comprising 73 SNPs and 22 Indels among the 1619 SNP sites of 16 *BnP5CR2* nucleotide sequences, and 12 polymorphic sites comprising 10 SNPs and two Indels among the 678 sites of the 16 CDS population. The sequence information, including DNA sequences of CDS, is listed in Supplementary Files S1 and S2. The locations with polymorphic differences among 16 *BnP5CR2* and its CDS are shown in Figure 2.

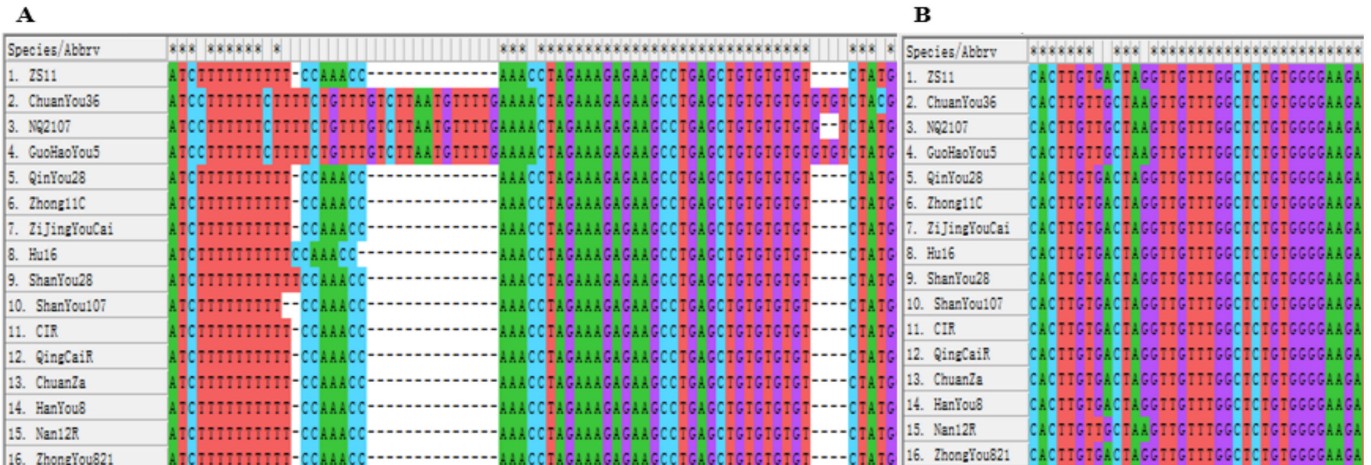

**Figure 2.** Sequence alignment of *BnP5CR2*. (**A**) Alignment based on gene. (**B**) Alignment based on CDS.

*3.3. Evolutionary Tree of BnP5CR2*

A phylogenetic tree was constructed using data from the previous step. The tree with the highest log likelihood is shown in Figure 3A,B. The proportion of population genetic variation explained by CDS was greater than that explained by *BnP5CR2*, considering the test value and phenotypic data (i.e., the stem lesion lengths after 14 days of *Sclerotinia* infection of the cultivars).

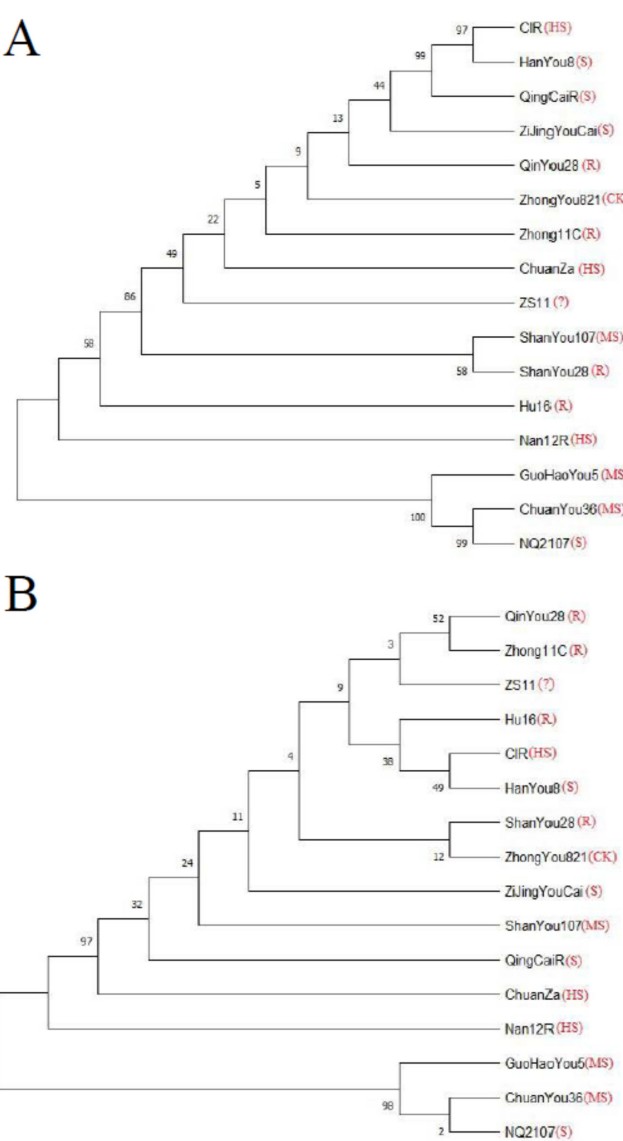

**Figure 3.** The evolutionary tree of *BnP5CR2* gene; (**A**) based on gene sequence; (**B**) based on CDS. "?" means that the disease resistance of Z11 is unknown.

*3.4. BnP5CR2-CDS-Haplotype Diversity and Network*

This analysis involved 16 nucleotide sequences, and the tree with the highest log likelihood is shown. Initial trees for the heuristic search were obtained automatically by applying Neighbor-Joining and BioNJ algorithms to a matrix of pairwise distances estimated using the Tamura-Nei model and then selecting the topology with the superior log likelihood value. A total of 10 mutations in the final data set and six haplotypes were observed in 16 CDS of *BnP5CR2* (Figure 4). The value of Hd was 0.617. The variance of the haplotype diversity was 0.018, and the standard deviation of haplotype diversity was 0.135. A total of 10 cultivars possessed the H1 haplotype: ZS11, QinYou28, ZiJingYouCai, Hu16, ShanYou28, ShanYou107, CIR, QingCaiR, ChuanZa, and HanYou8. The H2 Haplotype was present in ChuanYou36 and GuoHaoYou5; the H3 haplotype occurred in NQ2107; H4 haplotype occurred in Zhong11C; H5 haplotype occurred in Nan12R, and H6 haplotype was present in ZhongYou821. Some haplotypes appeared to be exclusive to a particular cultivar.

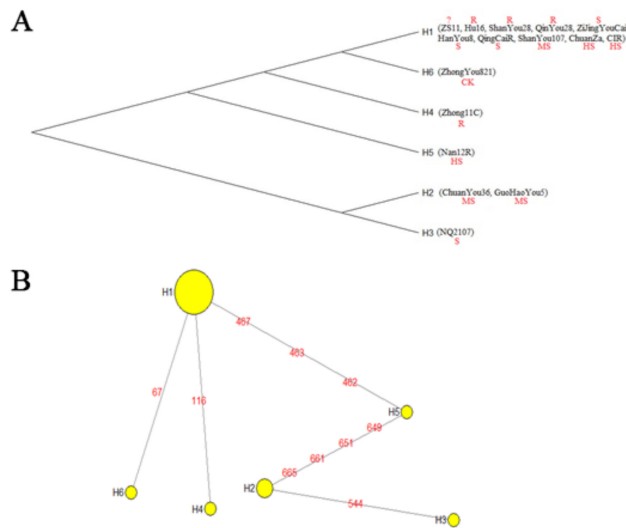

**Figure 4.** *BnP5CR2* gene-CDS-haplotype diversity. (**A**) Haplotype network. (**B**) Haplotype phylogeny. Note: (**A**) Red color is resistance rating corresponding to each cultivar. "?" means that the disease resistance of Z11 is unknown. (**B**) The size of the yellow nodes is proportional to frequencies. The red value displays mutated positions.

### 3.5. Expression Features of the BnP5CR2 Gene

Zhongyou 821, the disease-resistant cultivar of *B. napus*, is often used as a control check as resistance in disease resistance tests [30], and Nan12R was chosen at random as the disease-susceptible cultivar to *S. sclerotiorum*. The dynamic expression features of the *BnP5CR2* gene in cultivars Nan12R and ZhongYou821 are shown in Figure 5A,B, respectively. In total, the expression level of the *BnP5CR2* gene in the disease-resistant cultivars was significantly higher than that in the disease-susceptible cultivars, and the expression of the gene in the disease-resistant cultivars was significantly different versus time 0. The expression peak of this gene appeared 24 h after inoculation in cultivar Nan12R, while the expression level in ZhongYou821 changed significantly at 6 h. Comparing the two cultivars, the relative expression level of *BnP5CR2* was very low in Nan12R; in contrast, the expression of *BnP5CR2* was rapidly induced in ZhongYou821 by *S. sclerotiorum*. The relative expression level of *BnP5CR2* mRNA was significantly different between Nan12R and ZhongYou821 under 6 h of pathogen stress ($p < 0.01$).

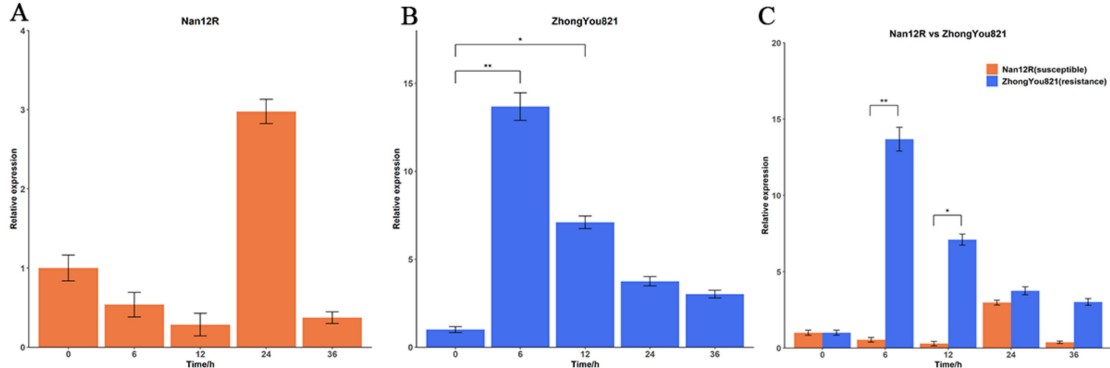

**Figure 5.** Expression patterns of *BnP5CR2* mRNA at 0 h, 6 h, 12 h, 24 h, 36 h treatments under pathogen stress. (**A**). Nan12R; (**B**) ZhongYou821; (**C**) Nan12R vs ZhongYou821. Note: * Significant difference at $p < 0.05$; ** significant difference at $p < 0.01$.

## 4. Discussion

In recent years, breeding disease-resistant cultivars have become one of the most important goals of *B. napus* cultivation. However, no immune cultivars were found in the screening with *S. sclerotiorum* antigen. The physiological and biochemical processes of plant disease resistance, quantitative traits controlled by multiple genes, are the result of interactions and co-regulation between genes and between genes and the environment. Different plants may resist adverse environmental conditions through multiple metabolic pathways under specific ecological conditions.

Many studies on *S. sclerotiorum* resistance have identified the resistance-related enzymes that are involved in this process by identifying the changes in defense-related gene expression profiles or enzyme profiles in *B. napus* under treatments with pathogens such as oxalate oxidase, superoxide dismutase, polyphenol oxidase, peroxidase, and phenylalanine ammonia lyase [31,32]. There are also studies in which no significant relationship was detected between *S. sclerotiorum* resistance and the expression of OXO, Cu/Zn SOD, PR2 and chitinase PR3 [33]. In recent years there has been an increasing research focus on omics concerning *S. sclerotiorum* infection; these studies have provided information concerning the mechanism of resistance. The candidate gene networks such as *P5CR*, cellulose synthase family protein, and glutamate synthase were identified as being involved in resistance to *S. sclerotiorum* in soybeans [1]. The upregulated *S. sclerotiorum* genes in chickpea lines revealed GO enrichment including oxidation–reduction processes, metabolic processes, carbohydrate metabolic processes, response to stimulus, and signal transduction [34]. The enrichment analysis showed that many of the hormone-signaling pathways in *B. napus* were activated under treatment with *S. sclerotiorum* post-inoculation [35]. The *BnPGIP*-overexpression lines of *B. napus* showed significantly increased *S. sclerotiorum* resistance, displaying a lower accumulation of $H_2O_2$ and over-expressed defense response genes within the salicylic acid and jasmonic acid/ethylene pathways [36]. The study of the *P5CR* gene may offer a fresh perspective on disease resistance in *B. napus*. In the post-genome era, the development of high-throughput sequencing technology has greatly facilitated the research on genetic polymorphism, and haplotype-based association tests may be better than those based on single SNPs (Figure 4). The diversity of gene-CDS-haplotype (gcHaps) is of particular importance for genetic variants in plant genetic research, Zhang et al. shows that there are $226 \pm 390$ gcHaps per gene in rice, resulting in the diversity of the same gene among different materials [19]. However, so far, few studies have been reported in this area. A total of six gcHaps were found based on 16 *BnP5CR2* genes in this study. H1 is an ancient haplotype with ten shared cultivars. The expression level of the *BnP5CR2* gene in Nan12R was lower than that of Zhongyou 821 during the entire cycle of pathogen stress; there was no significant difference at 24 h, suggesting that the response to *S. sclerotiorum* of resistant cultivars happens earlier than in susceptible cultivars, also suggesting that the H6 and H5 haplotypes may be associated with resistance and susceptibility to *S. sclerotiorum*, respectively.

## 5. Conclusions

Through polymorphism analyses of *BnP5CR2*-CDS-SNP and *BnP5CR2*-CDS-haplotypes that were associated with different levels of resistance to *S. sclerotiorum*, the results of the present study indicate that gene-CDS-haplotype diversity may have greater power than SNPs for the detection of causal genes for quantitative traits. The *BnP5CR2* gene may be involved in the response to *S. sclerotiorum* infection in *B. napus*, and thus the present study provides reference information for the elucidation of the pathogenesis of *Sclerotinia* toward the genus *Brassica* and future studies on the relationship between *P5CR* and disease resistance.

**Supplementary Materials:** The following supporting information can be downloaded at: https://www.mdpi.com/article/10.3390/agronomy12122956/s1. File S1 and File S2.

**Author Contributions:** Conceptualization, Y.Z.; methodology, Y.Z. and X.Z.; formal analysis, Y.W. and D.W.; resources, Y.Z.; data curation, X.S. and D.Q.; writing—original draft preparation, Y.Z.;

writing—review and editing, Y.Z. and D.Q.; supervision, Y.Z.; funding acquisition, Y.Z. and X.Z. All authors have read and agreed to the published version of the manuscript.

**Funding:** This research was funded by Shaanxi Provincial Department of Science and Technology Key Research and Development Program, grant number 2020NY-068, Scientific Research Program Funded by Shaanxi University of Technology, grant number SLGRCQD2115.

**Data Availability Statement:** The datasets used and analyzed during the current study are available from the corresponding author.

**Conflicts of Interest:** The authors declare no conflict of interest.

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
