# Peer review of "SNP and Haplotype Variability in the BnP5CR2 Gene and Association with Resistance and Susceptible Cultivars for Sclerotinia sclerotiorum in Brassica napus"

_agronomy, doi:10.3390/agronomy12122956_

Round 1
Reviewer 1 Report
General comments:
A minor review of English grammar and style are required as there is awkward language used in various parts of the manuscript, particularly in relation to tense and pluralization.
Please review formatting on species names, there are multiple instances where species names are presented in italics, while examples on lines 21, 54, 61, 65 and 70 show species names with incorrect formatting.
There are sections in the results that should be in the methods section. Examples: Lines 150-154, 207-211. The studies methodology should be reproducible without having to refer to the results or discussion.
Lines 19 and 22: sclerotium are the hardened fruiting bodies of some species of fungi, not just S. sclerotiorum. The authors did not screen for fungal bodies as part of their study. Please amend to clarify that they were looking at the pathogen Sclerotinia sclerotiorum.
Line 25: It is unclear what HD refers to here, this is the first mention of this term in the manuscript.
Line 42: Please correct the typo “clon”
Line 46: It is unclear what the authors are referring to in regards to “physiologic conditions”
Throughout the introduction authors first initials are included with the references, this is unusual. Please refer to the style guide.
Line 83: Further details are needed in the materials and methods. The study cannot be reproduced in it’s current form based on the authors descriptions. The authors need to indicate what type of materials they were studying (list cultivars) and provide methodology of their rating system. The authors do not describes how ratings were undertaken and analyzed, making the work non-reproducible. Please define the rating scale as well as the thresholds for resistant or susceptible varieties as this is a quantitative trait.
The authors need to describe the toothpick inoculation method in full, provide a reference, plant staging, growth conditions, and plant tissue inoculated. Again, this is needed to ensure the study is reproducible. The first reference to this methodology appears in figure 3 when it should be in the methods.
Line 97: Please correct grammar, lines typically do not start with a preposition in English.
Line 101: What is a BioSpin gel? This is unclear, clarify and provide a source.
Line 105: What type of sequencing was conducted? This is needed for reproducibility.
Line 115: The description of the phylogenetic analysis is lacking, it is unclear what steps were taken to ensure statistically sound results. Please list the number of replicates and provide bootstrapping details for the ML tree analyses. Again, needed to ensure reproducibility.
Line 132: What are “third-to-last leaves”? Are these at a given node? Last to what?
Figure 2: How were these aligned? Again, methods are lacking.
Please provide tree support values.
Author Response
Thank you very much for considering our manuscript (Agronomy Manuscript #:2005571) and for the reviewers’ comments and suggestions. The manuscript has been extensively and carefully revised according to the reviewers’ suggestions. The modified parts have been marked by red color in the revised manuscript, and the explanations for the specific comments of reviewers are shown below. Finally, we have uploaded a 'Response to Reviewer Comments'.

Reviewer 2 Report
Agronomy Manuscript #: 2005571
Authors: Y. Zhang et al.
Title: SNP and haplotype variability in the C09-BnaP5CR gene and association with resistance and susceptibility accessions for Sclerotinia sclerotiorum in Brassica napus
The authors have conducted a screen of 15 different cultivars (“accessions”) of Brassica napus (common name canola, “rapeseed”, or “rape”) that have more or less resistance to the important fungal pathogen Sclerotinia and then used genomic, DNA sequence analysis, and RNA expression methods to study a candidate resistance gene in several of the cultivars. The overall concept and approach is sound, and has yielded what seems valid information. Unfortunately, there are a number of key issues that need to be addressed. There are several open questions about which samples were included or not (and why) for RNA expression analysis, there is missing significant amounts of key information in the Methods section, and there is need of significant rewriting for clarity and readability in numerous sections. My key issues/concerns are listed below. Thus, the manuscript is not in a state that is ready for publication at this point, in my opinion.
I have several general concerns that relate to the entire manuscript in both writing and scientific clarity and soundness. These include:
1) First, the old traditional name of “rape” or “rapeseed” is a legitimate name, but for obvious reasons it has often been replace with the common name Canola. Now, that name can have specific connotations for low acid oil production and related to Canadian derived Brassica napus cultivars, and might be less appropriate for a range of Brassica napus cultivars that have not been selected for low acid oil content. I would encourage the authors to consider using Canola as the common name, if that is in fact fitting.
2) The proposed gene names for “C09-BnaP5CR” and “A10-BnaP4CR” are not compliant with plant gene naming conventions. Having been involved in naming several plant gene names that have been published/accepted, I am not aware of any gene names that include the Chromosome number in the gene name. Please select gene names that follow plant gene naming conventions, per accepted methods:
Arabidopsis gene naming convention: https://doi.org/10.1126/science.282.5389.662
Another example, from Merchantia: https://doi.org/10.1093/pcp/pcv193
3) There are several aspects of Methods and Materials that are completely missing and are essential and need to be cited with appropriate references, including the following key methods:
(1) details about how the plants were grown for genomic DNA isolation (greenhouse, field, etc…?).
(2) how old were plants for genomic DNA isolation? “Young leaves” (L95) is not sufficient
(3) where were all 16 of the B. napus cultivars obtained from (cite sources/references)?
(4) what method of DNA sequencing (L106) was used (Sanger? Illumina? Pacific Biosciences? Other?)
(5) how many days old were “seedlings” used for the gene expression experiments (L132)? And, were these field or greenhouse grown plants?
(6) must cite references for RNA extraction for RT-qPCR.
(7) description for the RT-qPCR method is incomplete. It must include more information about the qPCR method. What fluorescence method was used (SYBR? PacMan? Other?) Details about the instrumentation and software used for quantification is needed. And, were standard follow up melt curves and amplification “efficiency” data obtained to confirm the PCR was correct and only one amplified PCR product? These are standard for presenting RT-qPCR data to have confidence in the presented data, and need to be included here.
4) Some of the cultivars included in the sequence data (Figure 2 and elsewhere) were not included in the fungal infectivity data (Table 3). For example, ZS11 is the reference genome for DNA sequence yet it is not included in the fungal infection susceptibility data (Table 3). Seems it should be included in Table 3 or an explanation why not.
5) The two cultivars selected for more extensive analysis to compare gene expression / RNA levels by RT-qPCR between a resistant and a susceptible culture were Nan12R (susceptible) and Zhongyou821 (resistant). Yet, according to their data (Table 3), these are neither the most resistant nor most sensitive strains. Those would be H16 (most resistant) and CIR (most sensitive). On L239 the authors mention that Nan12R was selected randomly as a susceptible cultivar and Zhongyou821 because it has “partial resistance”. It is not clear to me that that strategy is the wisest if their goal is to identify how resistant cultivars differ from susceptible. Perhaps there are other variables or factors that impact this decision, and if so the authors should explain these more clearly.
6) The p-value statistical data shown in Figure 5 seems inconsistent in what is presented as significantly different and what is not. For example, and based on the provided averages and SEM error bars, in Zhongyou821 the 6 hours versus 12, 24 and 48 seem like they too would be significantly different. Similarly, for Nan12R, is the 24 hour data versus the other time points (0, 6, 12, and 24 hours) NOT significantly different? It seems unlikely considering how dramatically different the averages are and in light of the relatively small SEM error bars. And, similar again, for Figure 5C, it seems likely at 36 and possibly even 24 hours. There might be significant differences between Nan12R and Zhongyou821 for C09-BnaP5CR RNA levels?
6) In many locations, there are sentences/sections that need more reference citations, some of these are listed below in my detailed comments.
7) It seems a concluding figure that would correlate the haplotypes with which cultivars (accessions) and level of resistance. These data are discussed on L214 – L218, but a concluding figure that also shows fungal resistance/susceptibility level would be helpful.
Detailed/Specific Comments:
Lines 15 and 16: as mentioned above, the gene names “C09-BnaP5CR” and “A10-BnaP4CR” do not follow conventional plant gene naming methods. Further, gene and RNA names should be italicized. Throughout the manuscript sometimes the gene names are italicized while other times they are not. Please be consistent.
Line 18: as mentioned above, would the name Canola be a better option?
Line 25: Define “Hd” (haplotype diversity)
Line 27: Seems wording “…were present in …” would be better than “…were possessed…”.
Line 27: mention that Nan12R and Zhongyou821 are “cultivars” of Brassica napus. Related, throughout the manuscript the terms “cultivar”, “accession”, “sample”, and “breed” are used to describe these different “genetic lines.” I suggest using “cultivar” throughout the manuscript to avoid confusion.
Lines 29-32: somewhat of an awkward sentence, and it is not clear how “significant” versus “highly significant” differ statistically. Suggest rewriting for clarity.
Lines 38-41: long and awkward sentence. Rewrite for clarity. Further, I would change “…which may infect …” to read, “…which can infect…”
Lines 45-46: awkward wording in this sentence around “in the Eukaryote, under …” Reword.
Line 49: awkward wording in this sentence around “…an indispensible substance in protein…” Reword. Perhaps, “…an essential amino acid…”?
Line 54: Genus species need to be italicized, such as for Arabidopsis thaliana, and others throughout the manuscript.
Lines 85-86: awkward wording in this sentence around “…which responsible by Hanzhong …” Reword.
Line 93: After toothpicks were incubated for 7 days, then what happened? Stored or used immediately for infection of plants?
Line 95: How old were ”young leaves” in days since planting and where/how were plants grown (field, greenhouse)? There is key information missing.
Line 97: I can guess, but it is not clear what a “two couple primers” are. Reword for clarity.
Line 102, Table 1: On my file, the font for much of Table 1 was very odd/unusual. This might be a computer technical issue. Not sure.
Line 110: Delete “(Kumar et al., 2018)”, for the citation number [21] is sufficient.
Line 111: awkward wording in this sentence around “…were got and saved …” Reword.
Line 117: Delete “(Tamura et al., 1993)”, for the citation number [22] is sufficient.
Line 118: Should be, “Neighbor-Joining”.
Line 132: How old were ”seedlings” and where/how were plants grown (field, greenhouse)? There is key information missing.
Lines 134-137: A citation reference is needed for this method section.
Line 138: awkward wording in this sentence, “Finally, the leaves were taken out.” Reword to be clear meaning/purpose.
Line 140: awkward wording in this sentence, “Doing a three times biological repeat” Reword.
Lines 140-145: Citation references are needed for the RNA isolation and the RT-qPCR, along with a much more complete description of method for the RT-qPCR, as mentioned above.
Lines 151-153: This is more of Methods section than Results.
Line 153: This states the infection method was done “subsequent to continuous inoculations…” Does this mean the infections were allowed to go over three years? This can not be for a single plant, as an annual. A more clear explanation and wording for what was done is needed.
Line 154: Numbers under 10, such as “3” should be written out in word form, “three.” There are other examples of this throughout the manuscript.
Figure 1D: The picture of the inoculated plates are distorted, from what I see. A better picture is needed.
Table 3: The way the data are presented is very unclear and hard to easily see. Separating different datum points by “/” is not a good format and to then included the average +/- SD in same list is also not clear. Reformat Table to more clearly show data.
Table 3: Also, why is ZS11, reference cultivar, not included in the infectivity data?
Line 172: Since all of these variants are part of the same “locus”, there is only one “locus” (not “loci = plural”). There are 16 different versions / orthologs, but not 16 different loci.
Line 186: Should be “using data from Figure 2”, not previous step, which sounds like a Method.
Figure 3: The names of the top cultivars are cut off from the figure (ChuanZa for 3A and CIR for 3B). Reformat figure to include these.
Lines 207-211: These section needs to cite Figure 4A and 4B, somewhere. All figures should be cited in the Results text.
Line 217: add “haplotype” in front of both H4 and H6, to be consistent with above sentences.
Line 214-218: As mentioned above, I feel a simple figure that shows information listed in sentence, along with resistance/susceptibility levels might be a better way to show these key data, especially for a concluding figure.
Figure 4. Place panel “B” heading on upper left side of panel (versus upper right) side, to be consistent with other figures.
Lines 238-239: As mentioned above, the selection of Zhongyou821 and Nan12R for detailed RT-qPCR analysis is not the clear, best options for a resistant versus susceptible comparison among these cultivars. Better explanation as why the two selected were used over the two those that are more “resistant” versus more “susceptible”.
Line 241: Should be citations to “Figure 5A and 5B”, not Figure 4A and 4B.
Line 243: Citation for Figure 5C is needed.
Line 244: A statement about a difference “before” inoculation can not be made since there is not a “before” inoculation. Versus time 0 yes, but not “before” inoculation.
Figure 5: Figure legend needs information about the “n” number and what error bars mean, SEM? SD? Not clear.
Lines 271-274: Make it clear that these are genes.
Line 276: A reference/citation must be added since the sentence says, “studies have provided…” Cite those studies.
Line 281: Since there are so very many plant hormone-signaling pathways, it is unlikely “most of the hormone-signaling pathways” are affected. I suggest change wording to read, “…many of the hormone-signaling pathways…”.
Line 282: Seems wording “activated” versus “enriched” would be better.
Line 292: I would start a new sentence with, “However, so far, …”
Line 293: change wording from “made on this field.” to “reported in this area.”
Line 303: As mentioned above, twice, including Lines 214-218, a simple figure that shows information about what cultivars have what haplotypes plus the resistance/susceptibility levels of those cultivars might be a better way to show an overall conclusion from these data.
Author Response

(The authors gave the same response as above.)

Reviewer 3 Report
This article submitted for revision presents an interesting research important for the resistant breeding in Brassica napus against Sclerotinia sclerotiorum. The examinations are properly designed and obtained results are well presented. This research provides reference information for future studies on the relationship between P5CR and disease resistance.
Author Response
Thank you for your suggestions. We have carefully checked and corrected the language spelling.
Round 2
Reviewer 2 Report
Agronomy Manuscript #: 2005571
Authors: Y. Zhang et al.
Title (new): SNP and haplotype variability in the BnaP5CR gene and association with resistance and susceptibility cultivars for Sclerotinia sclerotiorum in Brassica napus
This is a re-review of the manuscript after the authors re-submitted this work. The authors did respond to and addressed the majority of my concerns, which is good. But, as I see it, there are still some issues that need to be resolved before publication. These are listed below.
First, there were some major changes in figure and table numbering and location. Some of which I do not think helped. First, as I see it, Figure 1 and the new Table 1 (which was the old Table 3, but improved) should be put back into the Results Section, for they are results, and not moved to the Methods Section. Thus, I suggest the “new” Table 1 should be moved back to being Table 3 and Figure 1, stays this number, but should be moved back into the Results Section, and with the told text section (with suggested edits) put back into the Results.
These changes would move “new” Table 2 to be Table 1 and the “new” Table 3 to be Table 2.
Second, I feel the new version of Table 3 (old numbering and my suggested number from above), which helps readers to understand. However, the way it showed up and printed on my computer resulted formatting issues with some of the cultivar names, such that they are split over several lines. For example, “ShanYou28”, as I see it, is “ShanYou” on one line and “28” on line below.
Also related to Table 3 with Lesion-Lengths and Resistance Ratings, there are inconsistencies in the “rate” labeling. The label “MS” (defined in footnote as “moderately susceptible”) seems the wrong labeling for the term “moderate” implies lesser or mildly. Yet, those cultivars labeled “MS” are in fact more susceptible than those labeled “susceptible”, according to the Lesion-Length values. Furthermore, ZhongYou 821 cultivar’s Resistance Rating is described as “CK” (aka, control check) for its resistance. That description does not relate to a Resistance Rating, but instead to how it is being used. Later (Line 292) ZhongYou 821 said to be “partially resistant”. If that is the case, then that should be indicated in this Table. However, according to the Lesion-Length, it seems ZhongYou 821 would be very close to the “Susceptible” (S) level. Again, there are inconsistencies with the Resistance Ratings shown in this Table, and the terminology being used. It should be corrected to be consistent.
The reason this is important is because ZhongYou 821 is later in the Results (Line 292 and Line 298) and Discussion (Lines 348-353) is being use as an example of a “Resistant” cultivar, when according to Table 3 it is on the boarder of a resistant or susceptible strain. This is an issue I mentioned previously, and asked why the authors do not test a truly resistant strain such as Hu16, in addition to the ZhongYou 821 “control.” As I see it, this is still an important experiment that is needed.
Third, for Figure 5, the authors still do not indicate what the error bars represent in this figure, and they did not provide information about p-values (statistical significance) for some of the data, as requested. For example, it seems very, very likely that in Fig 5A, there is a significant difference between 24 hr and 36 hr time points for Bn5CR2 mRNA in Nan12R cultivar, yet this is not mentioned. Perhaps it is not significant (p > 0.05), but then the error bars would seem very misleading. Similar issue for other data in Fig 5A, Fig 5B and Fig 5C. Since this is one of the key figures with data claimed to be very important for how this gene/RNA impacts resistance, it seems knowing these observed differences are significant is critical.
Specific issues still to be resolved:
Lines 56-58: Some verb-tense issues in these sentences. Should be, “… gene was cloned …” and “…genes were cloned …”
Line 79: deleted on of the duplicated, “study”
Line 86: “Brassica” should be capitalized.
Line 89: An extra space between “…which identified…”
Line 93: Still a “Rape” leftover, after converting most of these to “canola”
Line 96: extra space at end, “retain ed”
Lines 108-109: Not sure what is going here, but it seems a title to a paper or a citation that was not properly formatted. It says, “…(“Rule of field-resistance …”, which it seems more likely it should be “Role of ..”. Again, not sure what is going here. Also, “Sclerotinia sclerotiorum” should be italicized.
Line 112-120??? As mentioned above, Figure 1 should be moved back into the Results, as it was original, for it is showing results.
Line 126: As mentioned above, Table 1 should be moved back into the Results, as it was original, for it is showing results. This then leads to changes in table numbering for Tables 1, 2 and 3.
Line 129 (Table 3, as I would number it and its footnote): (as described above) ZhongYou 821 is said to be “control check” (aka, CK) for its resistance. That description does not relate to a Resistance Rating. Later (Line 292) it said to be “partially resistant”. If that is the case, then that should be indicated in this Table. However, according to the Lesion-Length, it seems ZhongYou 821 would be as close to being a “Susceptible” cultivar as a “Resistant” cultivar.
Table 3 continued: (as described above) The label “MS” (defined in footnote as “moderately susceptible”) seems the wrong labeling for the term “moderate” implies lesser or mildly. Yet, those cultivars labeled “MS” are in fact more susceptible than those labeled “susceptible”, according to the Lesion-Length values. This terminology is not consistent, and should be redone.
Lines 142-143: This is an awkward sentence that needs to be corrected/rewritten.
Line 177: Replace “when” with “that”
Line 180: Need to cite what company/source was used for TRIzol.
Line 187. Reference citation as well as source for SYBR Green and the SYBR Green method for needed, for there are many different sources and ways this can be done. Further, need to indicate what real-time PCR machine/equipment was used as well as fluorescence capture software, for these also vary considerably.
Lines 248-259 (and Results Section 3.3): Still need to cite Figure 4A and 4B somewhere in this section, for as is Figure 4 is not cited anywhere in the manuscript.
Figure 4: I like the addition of the cultivars for which the different haplotypes are present, which helps. However, please explain the “?” above ZS11 in Figure 4A. This likely comes back to the fact that this cultivar is include yet there are no “resistance” data provided, a concern I mentioned previously.
Line 292: (as mentioned above, twice) For Zhongyou821, which is frequently a control used in many studies, it is described as being “partially resistant”, yet in “new” Table 1 (what I would change to Table 3), ZhongYou821 is described as “CK” (for control check).
Lines 297 and 298: There is one “susceptible” cultivar (Nan12R) and one “partially resistant” cultivar (Zhongyou 821), so these should be listed as “the disease-susceptible cultivar” (not cultivars) and similarly singular “cultivar” for “disease-resistant”, unless they have and included additional data from more cultivars.
Line 313: (as mentioned above) What error bars is showing needs to be mentioned. SD? SEM?
Line 325: The names “OXO”, “SOD”, etc… are indicated as being “pathogens”, but these abbreviations seem more likely to be gene names, as is mentioned for similar names on Line 327.
Line 325: In an attempt to look up and read the cited reference for the above issue about “gene names” in the Liu et al., 2015 reference, I was not able to find this reference anywhere nor did the provided “doi.org” number in the Bibliography/Reference List work. Please check on this reference in the Bibliography to correct it.
Lines 348-353: A citation to Figure 4 is needed here.
Lines 402-403: (as mentioned above, Line 325) I was not able find this reference nor did the provided “doi.org” number work. Now, I did not check any of the other references, but it would be good to confirm that all the references and “doi.org” numbers are correct.
Author Response
Thank you for your suggestions. We have revised all the issues in the manuscript. Please check them.
